# An Investigation of Left Ventricular Valve Disorders and the Mechano-Electric Feedback Using a Synergistic Lumped Parameter Cardiovascular Numerical Model

**DOI:** 10.3390/bioengineering9090454

**Published:** 2022-09-08

**Authors:** Nicholas Pearce, Eun-jin Kim

**Affiliations:** Center for Fluids and Complex Systems (FSC), Faculty of Engineering, Environment and Computing, Coventry University, Coventry CV1 5FB, UK

**Keywords:** non-linear dynamics, multiscale model, cardiac cycle, mechano-electric feedback, lumped-parameter model

## Abstract

Cardiac diseases and failure make up one of largest contributions to global mortality and significantly detriment the quality of life for millions of others. Disorders in the valves of the left ventricle are a prominent example of heart disease, with prolapse, regurgitation, and stenoses—the three main valve disorders. It is widely known that mitral valve prolapse increases the susceptibility to cardiac arrhythmia. Here, we investigate stenoses and regurgitation of the mitral and aortic valves in the left ventricle using a synergistic low-order numerical model. The model synergy derives from the incorporation of the mechanical, chemical, and electrical elements. As an alternative framework to the time-varying elastance (TVE) method, it allows feedback mechanisms at work in the heart to be considered. The TVE model imposes the ventricular pressure–volume relationship using a periodic function rather than calculating it consistently. Using our synergistic approach, the effects of valve disorders on the mechano-electric-feedback (MEF) are investigated. The MEF is the influence of cellular mechanics on the electrical activity, and significantly contributes to the generation of arrhythmia. We further investigate stenoses and regurgitation of the mitral and aortic valves and their relationship with the MEF and generation of arrhythmia. Mitral valve stenosis is found to increase the sensitivity to arrhythmia-stimulating systolic stretch, and reduces the sensitivity to diastolic stretch. Aortic valve stenosis does not change the sensitivity to arrhythmia-stimulating stretch, and regurgitation reduces it. A key result is found when valve regurgitation is accompanied by diastolic stretch. In the presence of MEF disorder, ectopic beats become far more frequent when accompanied by valve regurgitation. Therefore, arrhythmia resulting from a disorder in the MEF will be more severe when valve regurgitation is present.

## 1. Introduction

Out of the many cardiovascular diseases plaguing millions of people throughout the world, disorders in the heart’s valves make up a significant proportion. These disorders frequently lead to death or morbidity, particularly in the ageing population [1]. Disorders in the left ventricle valves are more numerous than those in the right ventricle, with disease in the aortic valve making up the largest proportion of valvular deaths [2]. The three main disorders affecting the heart valves are prolapse, stenosis and regurgitation. Further, these pathologies can increase susceptibility to arrhythmia in the atrium and whole heart. It is known, for example that mitral valve prolapse may excite the electrical dynamics of the heart leading to cardiac arrhythmia [3]. This is thought to be caused by the increased stretching of the valve leaflets during systole, which in turn excites the electrical messaging in the ventricle via a mechanism called the mechano-electric feedback (MEF) [4]. Additional to the resulting arrhythmia, sudden cardiac death may also occur [3]. Valve prolapse may also cause regurgitation, which increases myocardial load and hence, stretching of the cardiac muscles [5]. Studies of valve patients that have recovered from arrhythmia as a complication frequently find fibrosis in the left-ventricular wall; evidence of excessive stretch [6]. Mitral valve stenosis is often associated with arrhythmia and is due to the excessive stretching of the left atrium [7].

Despite the heart’s apparent complexity, it remains a well-regulated organ and this is aided by many feedback mechanisms at work over multiple scales and domains; from the micro cellular scale of the myocytes to the macro organ level scale. The MEF is one of the main feedback mechanisms. Along with the electro-excitation coupling (ECC), these two mechanisms help maintain cardiac stability and synchronicity. As their names suggest, the MEF is the feedback of local mechanical cell stretch on the electrical dynamics, while the ECC operates in the opposite direction. For comprehensive reviews of the MEF see [8,9]. Whilst the MEF helps maintain synchronicity, it can have some peculiar consequences, particularly in the generation and termination of arrhythmia. Commitio-cordis, which is the mortal induction of fibrillation by an innocent impact to the chest is perhaps the most peculiar. Link et al. [10,11] conducted an illustrative set of clinical experiments studying how ventricular mechanical stretch can excite the cardiac electrical activity and induce arrhythmia. This is found to be highly dependent on the ECG timing, with only those cases in which stretching occurred during a certain window of the ECG initiating arrhythmia. A review of the recent clinical studies into mechanically induced electrophysiological behaviour is provided in [5]. Stretch-activated-channels (SACs) are the leading mechanism thought to be responsible for the MEF [9,12], though other mechanisms may also be involved. SACs are cells which open or close cellular ion channels in response to mechanical stretch and for the left ventricle, their response depends on the timing during the cardiac cycle [10,13,14]. Stimulating these channels can thereby change the character of the action potential: the electrical wave that causes cell contraction and relaxation. The action potential changes depend on the period in the cardiac cycle at which stretch is induced.

Computer and mathematical modelling is a powerful way to investigate complex systems as it allows for system visualisation, hypotheses and predictions to be examined at relatively low cost compared to experimental methods [15]. Computer models of the cardiovascular system vary in complexity, from simple zero dimensional (0D) and one dimensional (1D) models to full three dimensional (3D) models, some involving motion of structure. Numerical investigations of the heart valves likewise vary in complexity. 0D and 1D models involving the heart valves frequently use a ‘diode’ approach in which the dynamics of the valve are ignored and the direction of flow imposed similar to an electrical diode [16,17], while fully 3D examples most frequently use stiff geometry and studies using dynamic values [18] are rare. The 3D structural models provide flow field information and include interaction between the tissue structure and flow [18,19,20]. Complex numerical models involving cellular mechanics, electrophysiology, ion movements, and 3D models requiring detailed mathematical solution do not lend themselves to the demanding clinical environment due to their high cost in terms of computational load and time [21]. Numerical models of MEF likewise vary in complexity, with plenty of examples of relatively simple low-order 0D and 1D studies [12,22,23,24] and full cardiovascular system and 3D models [25,26,27].

In this study a 0D mathematical model of the left ventricle with valve stenoses and regurgitation is developed, by modifying the synergistic cardiovascular model by Kim and Capoccia [28]. The model consistently simulates the coupling of the mechanical, chemical and electrical functions of the myofiber on the micro-scale as well as the macro-scale organ activity. The consideration of different domains and scales gives it a synergistic quality, which is similar to Roy et al. [16] who control organ dynamics through the electrophysiology. Their haemodynamic activity is controlled using the time-varying-elastance (TVE) method however, in which the ventricular pressure–volume relationship is imposed using a periodic function [29,30] instead of calculating it consistently. The TVE paradigm has frequently been questioned [17,31] when used for cardiovascular modelling due to its empirical foundations and neglect of haemodynamic-regulating feedback mechanisms. Use of the adopted model [28] bypasses the need to use the TVE method due to its synergistic approach. The model has been validated as an alternative to the TVE method [28] and previously used for the study of dilated cardiomyopathy, left ventricular assist devices (LVADs) and MEF [32,33]. In [32], the model proved capable at reproducing known MEF effects consistent with previous findings by others, for example a prolonged action potential duration consistent with [34], and ectopic peaks in electrical patterns along with rapid oscillation consistent with the effect of SACs seen in [9,27,35]. The rest of the paper is organised as follows: the cardiovascular system model is described in Section 2; in Section 3, the results of using the model to simulate valve and MEF pathologies are described; in Section 4, these results are discussed; and in Section 5 a conclusion to this study is provided.

## 2. Materials and Methods

The cardiovascular system model is made of three parts presented in Section 2.1, Section 2.2, Section 2.3 and Section 2.4 as follows. First, in Section 2.1, the micro-scale sarcomere model is described. This is followed in Section 2.2, by the macro, organ scale dynamics model. The electro-chemical activity model is described in Section 2.3 and Section 2.4 describes the method for simulating valve pathologies. Table 1 gives the physiological meaning of the model variables. Table 2 and Table 3 provide the meaning and values of the model parameters used for a control case representing a ‘healthy’ pathology-free heart.

### 2.1. Micro-Scale Model

The micro-scale mechanical model by Kim and Capoccia [28] is based on the Bestel–Clement–Sorine (BCS) formulation [36,37] for myofiber dynamics. It is derived from the Hill–Maxwell rheological model of a myofiber in which the active contractile sarcomere is placed in series with another elastic element that simulates the active relaxation. These two elements are surrounded by connective tissues which is modelled by a third elastic element in parallel to the active and passive sarcomere parts. This parallel element stops the heart exceeding its limits [38]. The governing equations below describe the velocity $v_{c}=\frac{d\epsilon_{c}}{dt}$, strain $\epsilon_{c}$, stress $\tau_{c}$, and stiffness $k_{c}$ of the active contractile element, represented by the subscript ‘*c*’. Equation (Equation 5) below for $d_{0}\left(\epsilon_{c}\right)$ represents the Frank-Starling law for cell stretch.
(1)$$\begin{array}{cc} \frac{dv_{c}}{dt} & =-\chi \tau_{c}-\omega_{0}^{2}\epsilon_{c}-a\tau_{c}d_{0}\left(\epsilon_{c}\right)+b\left(\sqrt{\frac{V}{V_{0}}}-1\right), \end{array}$$
(2)$$\begin{array}{cc} \frac{d\epsilon_{c}}{dt} & =v_{c}, \end{array}$$
(3)$$\begin{array}{cc} \frac{d\tau_{c}}{dt} & =k_{c}v_{c}-(a_{l}|v_{c}\left|+|u|\right)\tau_{c}+\sigma_{0}u_{+}, \end{array}$$
(4)$$\begin{array}{cc} \frac{dk_{c}}{dt} & =-(a_{l}|v_{c}\left|+|u|\right)k_{c}+k_{0}u_{+}, \end{array}$$
(5)$$\begin{array}{cc} d_{0}\left(\epsilon \right) & =e^{-\beta_{0}{\left(\epsilon_{c}-0.1\right)}^{2}}. \end{array}$$

Here, the ‘+’ subscript means that only positive values of the preceding term are included, otherwise the term is 0. The first term $\chi \tau_{c}$ in Equation (Equation 1) represents a damping force, $\omega_{0}^{2}\epsilon_{c}$ represents a harmonic force, $a\tau_{c}d_{0}\left(\epsilon_{c}\right)$ is an active force, and $b\left(\sqrt{\frac{V}{V_{0}}}-1\right)$ is a passive force. *u* simulates the calcium bound Troponin-C concentration responsible for cell activation. The last term in Equations (Equation 3) and (Equation 4) represents the activation of the contractile force, while the term $\left(a_{l}|v_{c}|+|u|\right)$ models the deactivation. $\sigma_{0}$ and $k_{0}$ are constants representing the maximum cell stress and stiffness, respectively. The parallel stress element in the Hill–Maxwell model is represented here by $\sigma_{p}$ and evolves exponentially according to Equation (Equation 6) below.
(6)$$\begin{array}{cc} \sigma_{p} & =\frac{k_{2}}{k_{1}}\left[\exp\left(k_{1}\sqrt{V/V_{0}}\right)-1\right], \end{array}$$
in which $k_{1}$ and $k_{2}$ are constants for passive tension.

### 2.2. Macroscale Model

The micro-scale part of the model is coupled to the macro organ scale part through the left ventricle pressure $P_{v}$ according to Equation (Equation 7) below.
(7)$$\begin{array}{cc} P_{v} & =\gamma \frac{V_{0}}{V}\left[d_{0}\left(\epsilon_{c}\right)\tau_{c}+\sigma_{p}\right]. \end{array}$$
γ in Equation (Equation 7) is a constant representing the ventricular wall thickness to radius ratio. The ventricle volume *V* and circulation models evolve according to Equations (Equation 8)–(Equation 12) below, in which *m* is the aortic pressure, $F_{a}$ is the aortic flow, $P_{r}$ is the left atrial pressure, and $P_{s}$ is the systemic pressure.
(8)$$\begin{array}{c} \frac{dV}{dt}=\frac{U_{mi}\left(P_{r}-P_{v}\right)}{R_{m}}-\frac{U_{ao}\left(P_{v}-m\right)}{R_{a}}, \end{array}$$
(9)$$\begin{array}{c} \frac{dm}{dt}=-\frac{F_{a}}{C_{a}}+\frac{U_{ao}\left(P_{v}-m\right)}{C_{a}R_{a}}, \end{array}$$
(10)$$\begin{array}{c} \frac{dF_{a}}{dt}=\frac{m-P_{s}}{L_{s}}-\frac{R_{c}F_{a}}{L_{s}}, \end{array}$$
(11)$$\begin{array}{c} \frac{dP_{r}}{dt}=\frac{P_{s}-P_{r}}{C_{r}R_{s}}-\frac{U_{mi}\left(P_{r}-P_{v}\right)}{C_{r}R_{m}}, \end{array}$$
(12)$$\begin{array}{c} \frac{dP_{s}}{dt}=\frac{P_{r}-P_{s}}{C_{s}R_{s}}+\frac{F_{a}}{C_{s}}. \end{array}$$
$R_{m}$, $R_{a}$, $R_{c}$, and $R_{s}$ above are the aortic, atrial, characteristic, and systemic resistances, respectively. $C_{a}$, $C_{r}$, and $C_{s}$ are the aortic, left atrial, and systemic compliances. $U_{mi}$ and $U_{ao}$ are used for mitral and aortic disorders, respectively. They are described in Section 2.4 further below.

### 2.3. Electrico-Chemical Model

The electrophysiology is represented by the parameters *p* and *q* representing slow and fast electrical activity. Equations (Equation 13) and (Equation 14) below govern these variables, and Equation (Equation 15) describes the evolution of the chemical activity *u* which is proportional to the fast electrical activity variable *q*.
(13)$$\begin{array}{cc} \frac{dp}{dt} & =0.1(q-p+\mu_{1}\tau_{c}), \end{array}$$
(14)$$\begin{array}{cc} \frac{dq}{dt} & =10q\left(1-q^{2}\right)-10{\left(2\pi \right)}^{2}p+\mu_{2}V_{+}+10\cos\left(2\pi t\right), \end{array}$$
(15)$$\begin{array}{cc} u & =\alpha_{u}q. \end{array}$$

The constant parameters $\mu_{1}$ and $\mu_{2}$ model the MEF. Kim and Capoccia [32] describe $\mu_{1}$ as mimicking systolic mechanical stretch, whereas $\mu_{2}$ mimics diastolic stretch. Increasing either parameter increases the coupling between electrical and mechanical parts of the model, thereby increasing the stretch of the MEF. For further detail regarding these parameters and their effects, refer to [28,32,33]. The last term, 10cos(2πt) in Equation (Equation 14), simulates the heart rate (1 Hz) in the control case.

### 2.4. Valve Disorders

In the circulation equations above, two constant parameters $U_{mi}$ and $U_{ao}$ are used. $U_{mi}$ describes disorders of the mitral valve and $U_{ao}$ disorders of the aortic valve. In a ‘healthy’ heart without a disorder of either valve, both parameters are 1 when the preceding term is positive and 0 otherwise. For example, $U_{mi}$ is 1 when $\left(P_{r}-P_{v}\right)$ is positive and 0 otherwise. Likewise $U_{ao}$ is 1 when $\left(P_{v}-m\right)$ is positive and 0 otherwise. Different valves may be used to represent a valve pathology. For a stenosis, instead of $U_{mi}$ and $U_{ao}$ being 1 or 0, the 1 can be reduced, but 0 stays the same. To represent valve regurgitation, instead of $U_{mi}$ and $U_{ao}$ being 1 or 0, the 0 value can be increased depending on severity, but the value 1 must remain. Table 4 provides the values used for modelling different severities of each valve disorder. For a stenosis, the values represent the proportion of flow unobstructed. Therefore, the mild case represents a 30% flow restriction, the moderate case represents 50% restriction and the severe case represents 90% restriction. This covers a similar range of flow restriction to [16]. The values used for valve regurgitation are also identical to [16] who use a similar method to model valve disorders, and use the clinical definitions for regurgitation severity.

The equivalent electrical diagram for the complete model just described is shown in Appendix A section. The physiological meaning of the model variables is given in Table 1. The parameter values for the micro-scale mechanical and electrical models in Table 2 are those used by [28,33], and those for the macro-scale circulation model in Table 3 are provided by [39]. The system of Equations (Equation 1)–(Equation 15) are solved in Matlab R2021a (Mathworks) using a custom written fourth-order Runge–Kutta scheme and the accuracy of the numerical solution is checked by systematically reducing the time-step until differences between results become negligible.

## 3. Results

### 3.1. Mitral Valve Pathology

We first examine the effects of mitral valve pathologies without a dysfunction of the MEF. Figure 1a shows left ventricle pressure–volume loops (P-V) for the different severities of mitral valve stenosis recorded in Table 4. The loops can be seen to progressively shift downward and to the left as the severity worsens. The reduction in orifice area causes the end-systolic and diastolic pressures and volumes to fall, as does $P_{V}$ overall. Figure 1b,c show that as the stenosis worsens, the cardiac output and stroke volume reduce. The end-diastolic and systolic volumes (EDV and ESV) shift to the left; the EDV by 7 mL and the ESV by 1.5 mL. This is similar in degree to [16] who see a 5 mL shift in the EDV and 3 mL for the ESV between the mild and severe cases. The downward shift of the systolic and diastolic pressures is also similar. Figure 2a shows there is slight reduction in systemic pressure $P_{s}$, and Figure 2b shows there is an increase in the atrial pressure $P_{r}$.

Figure 3a shows the effects of mitral valve regurgitation on the left ventricle P-V loops for different severities of regurgitation. As the disease worsens, the systolic pressure falls as new flow is pumped back into the atrium, and the loops widen with an elimination of the isovolumic phases. Consequently, the stroke volume and cardiac output displayed in Figure 3b,c rise as the left-ventricle is enlarged [40]. The *x* axis of the figures give the severity of regurgition, where 1 denotes normal or control, 2 denotes mild, 3 denotes moderate, and 4 denotes severe. The model parameter values used to model each severity are provided in Table 4. The degree of changes displayed in Figure 3a agree well with [16,41].

Figure 4 shows how different severities of mitral valve regurgitation affect the model variables. As the severity of mitral valve regurgitation worsens, the systemic pressure falls further as more blood flows back through the mitral valve and less downstream. The atrial pressure $P_{r}$ consequently increases too due to the rise in blood back-flow. The aortic pressure *m* reduces for the same reason that the systemic pressure falls too.

### 3.2. Aortic Valve Pathology

Aortic valve pathologies make up the majority of valve disorders leading to mortalities. The effect of aortic valve stenoses corresponding to Table 4 on the left ventricular P-V loops is shown in Figure 5a. As the disease progresses, there is a slight increase in systolic pressure and end-systolic volume though some clinical reports suggest the change should be greater [42]. Figure 5b,c show, respectively, the corresponding reduction in stroke volume and cardiac output. Despite the severity of the stenosis, the effects for the aortic valve appear to be very mild compared to mitral valve stenosis. Not shown is that there is slight reduction in systemic blood pressure $P_{s}$ and an increase in atrial pressure $P_{r}$.

Figure 6a shows the effect of aortic valve regurgitation on the ventricle P-V loops. The stroke volume increases (Figure 6b) as the loops widen, as does cardiac output (Figure 6c). The widening of the loops is largely as a result of the end-diastolic volume (preload). The systolic pressure decreases, the diastolic pressure rises, and the isovolumetric phases become curved. As shown in Figure 7, the atrial pressure rises, whilst the systemic and aortic blood pressures fall. The change for aortic regurgitation found here more closely matches published literature [42,43].

### 3.3. Valve Motion

Before exploring the impact of valve disorders on the MEF, a brief examination of valve motion is made. To model valve motion, $U_{mi}$ and $U_{ao}$ are modified to produce a simple valve motion. For mitral valve opening then closing, the following is applied for the mitral valve
(16)$$\begin{array}{c} U_{mi}=\cos\theta ,\;P_{r}\geq P_{v} \end{array}$$
(17)$$\begin{array}{c} U_{mi}=1-\cos\theta ,\;P_{r}<P_{v} \end{array}$$

Likewise for the aortic valve,
(18)$$\begin{array}{c} U_{ao}=\cos\theta ,\;P_{v}\geq m \end{array}$$
(19)$$\begin{array}{c} U_{ao}=1-\cos\theta ,\;P_{v}<m \end{array}$$
where θ is a linear vector running from 90° to 0°. The angle θ may be thought of as being aligned with the direction of flow. The model above is far simpler than comparable low order models such as that used by [44] who also consider local flow effects around the valve, but it still allows for an exploration of valve opening and closing times. The time may be controlled by adjusting the speed with which θ completes its motion. It could also me used for an alternative way to model valve disorders too by adjusting the final value of θ. Figure 8 shows the effects of mitral valve opening and closing times on the pressure–volume loop. It can be seen that the closing time of the mitral valve has a much greater effect on the P-V loop than the opening of time and the longer the closing time, the greater the effect. The two plots in Figure 8 show the change in the variable $U_{mi}$. We refer the reader to Section 2.4 for an explanation of $U_{mi}$. The opening time of the valve does not appear to affect the closing time, but the closing time does affect the opening time of the next heart beat.

Figure 9 shows the effect of the aortic valve opening and closing times on the P-V loop with corresponding traces of the variable $U_{ao}$. From Figure 9, a similar conclusion can be made about the aortic valve as the mitral valve in that the valve closing time has a much greater impact on model behaviour than the opening time. To produce a similar model behaviour to the control-case (representing a typical ‘healthy’ person), the closing time of the left-ventricle valves must be short. This is similar to the observation by [44] who find that valve motion is brief and abrupt. Using their valve model, Korakianitis et al. [44] find that the entire valve opening and closing process takes 0.1 s. Using the cardiac model here, the closing time should make up a far smaller proportion of the 0.1 s. In the remainder of this study, valve motion is modelled as being instantaneous so that the focus remains on the MEF.

### 3.4. MEF

The response of MEF dysfunction to valve stenoses and regurgitation are now explored using the current model (without valve motion) by increasing the MEF parameters $\mu_{1}$ and $\mu_{2}$ with the addition of valve disorders. Kim and Capoccia [32] conduct a comparable study using a similar model without valve pathology. Appendix A section shows the effects of increasing $\mu_{1}$ for the control case in the model here without valve disorders. When $\mu_{1}=0.0128$ period doubling occurs, as shown in the top row of Appendix A. In the second row, when $\mu_{1}=0.0145$, 5 P-V loops can be seen which falls to 3 in the third row ($\mu_{1}=0.02$) and finally a single loop when $\mu_{1}=0.0268$. As noted above, the MEF parameter $\mu_{1}$ couples the slow activity variable *p* to ventricular stress $\tau_{c}$. As the stress is greatest during systole, $\mu_{1}$ mimics the effect of systolic mechanical stress (stretch) on the excitation wave. As $\mu_{1}$ is increased, the sensitivity to systolic stretch increases, causing the end-diastolic and systolic volumes to rise. The heart rate (hr) falls, showing that missed beats occur with increasing frequency as $\mu_{1}$ rises. The mean CO consequently reduces due to the reduction in hr. The slow electrical activity *p* increases faster to greater values, but falls more gradually which prolongs the repolarisation process causing a longer action potential (AP duration or APD); an effect also seen by Kim and Capoccia [32] and noted by others [12,23,45].

The MEF parameter $\mu_{2}$ couples the faster depolarisation electrical activity variable (represented here by *q*) to a rise in ventricular volume *V*, thereby mimicking the effect of diastolic mechanical stretch. A rise in $\mu_{2}$ increases the sensitivity to diastolic stretch. Appendix A section shows the effects of increasing $\mu_{2}$ as $\mu_{2}=\left[0.1530,0.1584,0.1718,0.36,1.8\right]$ on the control case ($\mu_{1}=0.0024$). Stretch during diastole causes SACs to open, allowing currents to flow into the cell which depolarises the action potential causing contraction [23,34]. Hence, the heart rate increases substantially, reaching 193 bpm when $\mu_{2}=1.8$. The stroke volume declines as the EDV reduces while the ESV rises. Despite the reduction in stroke, the CO rises; presumably due to the rapid rise in heart rate. Unlike systolic stretch, no periodic behaviour is seen. Instead, $\mu_{2}$ gives rise to complex P-V loops. The maximum value of the fast activity *q* shown in the last figure column can be seen to increase, whilst that of *p* reduces. This suggests that the AP depolarisation is stronger, and the repolarisation process weaker, hence the APD reduces. Unlike $\mu_{1}$, increasing $\mu_{2}$ does not produce similar periodic behaviour as is evident from the figures.

#### 3.4.1. Effect of $\mu_{1}$ ($\mu_{2}=0$) with Mitral Valve Pathology

The parameter $\mu_{1}$ is progressively increased ($\mu_{2}=0$) for the cases with a severe mitral valve stenosis or regurgitation, respectively, (refer to Table 4 for severity). In order to determine the effects of each pathology in the presence of MEF dysfunction, the average heart rate (hr), SV, and CO are compared with the control case. The value of $\mu_{1}$ at which periodic behaviour occurs is also compared. With a severe mitral valve disorder, instead of 2, 5, 3, and 1 P-V loops only 2, 4, 3, and 1 occur. Table 5 shows the results with the headings 2, 4 or 5, 3, and 1 representing two, four or five, three, and one P-V loops.

It can be seen that the sensitivity to systolic stretch is increased slightly with mitral valve stenosis, as the value of $\mu_{1}$ at which period doubling occurs is lower for stenosis. This is consistent with experimental studies of the mitral value that the rise in arrhythmia occurrences with mitral valve stenosis is due to higher atrial pressures, and an increase in intraventricular volume [46]. With a mitral valve regurgitation, period doubling occurs at a larger value of $\mu_{1}$ compared to the control case. Sensitivity to systolic stretch can therefore be concluded to reduce when mitral valve regurgitation is present. However, mitral valve pathology does not alter the effect of $\mu_{1}$. As $\mu_{1}$ increases, the SV, tends to increase but the hr and CO both tend to decline as they did for the control case. The reduction in CO in particular shows how the MEF can lead to cardiac problems and even death, as the pumping power of the heart diminishes.

#### 3.4.2. Effect of $\mu_{1}$ ($\mu_{2}=0$) with Aortic Valve Pathology

$\mu_{1}$ is now progressively increased with aortic valve pathologies in a similar fashion to the mitral valve. The results are tabulated in Table 6 along with the control case for comparison. Overall, the effects of $\mu_{1}$ are quite similar to the changes seen for the mitral valve and control cases. The end-diastolic and systolic volumes rise, the hr tends to fall as does the CO. Again the reductions in hr and CO show the damaging effects of the MEF. As $\mu_{1}$ at which period doubling first appears is slightly larger with an aortic valve stenosis or regurgitation than that without, a reduction in sensitivity to systolic stretch can be said to result from aortic valve disorders.

#### 3.4.3. Effect of $\mu_{2}$ ($\mu_{1}=0.0024$) with Mitral Valve Pathology

$\mu_{1}$ is returned to its control value $\mu_{1}=0.0024$, and $\mu_{2}$ is gradually increased with a severe mitral valve disorder (see Table 4 for severity). As noted in the control case above, $\mu_{2}$ does not produce periodic behaviour similar to $\mu_{1}$. The same values of $\mu_{2}$ are therefore compared between all cases. Specifically, $\mu_{2}$ is chosen to be $\mu_{2}=\left[0.1530,0.1584,0.1718,0.36,1.8\right]$ for all the cases with or without a valve disorder. The results are tabulated in Table 7 with $\mu_{2}$ increasing from left to right columns. The effects of $\mu_{2}$ on the ventricle are unchanged regardless of mitral valve disease. Ectopic beats appear and the heart rate increases as before. We note that the P-V loop in the control case without a valve pathology begins to bifurcate and show different loop shapes when $\mu_{2}=0.1516∼0.1517$. That is for any $\mu_{2}<0.1516$, the P-V loop appears normal and representative of a ‘healthy’ person. With a severe mitral valve stenosis, the P-V loops begin to bifurcate when $\mu_{2}\approx 0.1528$, hence mitral valve stenosis can be concluded to reduce the sensitivity to arrhythmia stimulating diastolic stretch. This is also evident in the hr, which does not increase to the same extent as the control case.

With a severe mitral valve regurgitation, the first appearance of bifurcation occurs when $\mu_{2}=0.1516∼0.1517$, which is identical to the case without valve pathology. Despite this result, the hr with regurgitation is increased immediately when diastolic stretch ($\mu_{2}\neq 0$) is introduced. Regurgitation also increases the hr at a greater rate compared to both the control case and mitral valve stenosis. The mechanism behind this result is evident in Equation (Equation 14). That is, mitral valve regurgitation produces a much wider diastolic stroke and a larger stroke volume. This increases the stretch (increase in volume) during this period, leading to a rise in ectopic beats and hr. Ventricular arrhythmia will therefore be more likely with valve regurgitation, with the greater the severity of regurgitation, the more severe the arrhythmia. This result should be expected when aortic valve regurgitation is present too as the stroke volume is larger again. These results are consistent with clinical studies which show that the severity of regurgitation is an indicator for the degree of complications experienced during cardiac arrhythmia [47,48,49].

#### 3.4.4. Effect of $\mu_{2}$ ($\mu_{1}=0.0024$) with Aortic Valve Pathology

With the parameter $\mu_{1}$ returned to its control value ($\mu_{1}=0.0024$), the value of $\mu_{2}$ is increased with the addition of aortic valve pathology in a similar way to the control case (as $\mu_{2}=\left[0.1530,0.1584,0.1718,0.36,1.80\right]$). Table 8 documents the results. A similar behaviour pattern to the case of the mitral value pathology appears as $\mu_{2}$ increases. The value at which bifurcation first appears is $\mu_{2}\approx 0.152$ which is similar to the control case hence, sensitivity to diastolic stretch is unaffected by aortic stenosis. With severe aortic regurgitation, the value at which bifurcation first appears is $\mu_{2}\approx 0.1515$. As expected, when aortic valve regurgitation is present and $\mu_{2}\neq 0$, ectopic beats become far more frequent, leading to a faster hr. Furthermore, the hr increases at a greater rate as $\mu_{2}$ rises, in agreement with findings above.

## 4. Discussion

Our aim was to develop a new model by extending a synergistic reduced-order mathematical model of the cardiovascular system [28] to include the effects of pathologies in the valves of the left-ventricle; the mitral valve and the aortic valve. A further aim was to see what effect if any, valve pathologies have on (disorders of) the mechano-electric physiology of the heart. In order to meet the latter aim, the popular time-varying elastance method of generating the pressure–volume relationship could not by applied. The TVE method uses a periodic function to generate the pressure–volume relationship rather than calculating it consistently, hence it cannot be used to simulate the feedback mechanisms which maintain cardiac stability, most notably the MEF. It has frequently been questioned both due to its empirical foundations and validity for cardiac modelling [17,31]. The synergistic model [28] develops the organ scale dynamics of pressure and volume from the micro-scale activity of the myocytes. Since it encompasses the mechanical, electrical and chemical domains it can be used to simulate the MEF. The model is modified to include stenoses and regurgitation in the heart’s mitral and aortic valves, modelling different severities of different valve disorders.

Mitral valve stenosis without a dysfunction of the MEF is found to cause a reduction in ventricular and systemic pressures and reduction in the cardiac output and stroke volume. The atrial pressure increases slightly. The EDV and ESV both reduce, shifting the P-V loop to the right. The extent of this reduction and shift is of similar order to [16] for a comparable range of severity. The results with mitral regurgitation show that the P-V loop widens, enlarging the ventricle, increasing the stroke volume, cardiac output, and myocardial load in agreement with medical reports [40]. The degree of the changes in the ventricle agrees well with [16,41]. The atrial pressure rises whilst the systemic blood pressure falls. For aortic valve stenosis, the ventricular pressure rises but this rise is very minor, even with 90% flow restriction. The stroke volume and cardiac output reduce but again, this reduction is very minor. Aortic valve regurgitation has a greater effect and agrees with with published literature [43]. Like the mitral valve, the stroke volume and cardiac output rise. This is due to an increase in the end-diastolic volume and slight decrease in end-systolic volume. The systemic blood pressure falls significantly during diastole, but the atrial pressure rises.

Dysfunction of the MEF is modelled by increasing the coupling between the mechanical part of the model and the electrical. Two parameters are used for this coupling; $\mu_{1}$ and $\mu_{2}$. The former mimics the effect of mechanical ventricular stretch during systole and the latter mimics the effect stretch during diastole. By increasing these parameters, dysfunctions of the MEF can be simulated and the effects on the cardiovascular system investigated. See [32,33] for their effects without valve disorders. A dysfunction of the MEF is investigated here with the addition of severe disorders in the mitral and aortic valves. Disorders in the mitral and aortic valves do not qualitatively change the overall effects of the MEF parameters. The parameter $\mu_{1}$ mimicking systolic stretch causes a reduction in the electrical activity, leading to missed heartbeats and a reduction in the pumping power of the heart, and is unaffected by disorders in the aortic and mitral valves. The parameter $\mu_{2}$ mimicking systolic stretch causes an increase in the electrical activity leading to ectopic beats and complex pressure–volume behaviour. This too is unaffected by disorders in the valves of the left ventricle.

Valve disorders do however have quantitative effects. Specifically, they change the sensitivity of the heart to arrhythmia-stimulating stretch. Sensitivity to systolic stretch is compared between cases by looking at the value of $\mu_{1}$ at which period doubling occurs. The lower the value of $\mu_{1}$ when periodic behaviour appears, the greater the sensitivity to systolic stretch. Mitral valve stenosis slightly increases the sensitivity to systolic stretch while regurgitation reduces it to a greater extent. Neither stenosis nor regurgitation affect the number and frequency of ectopic beats and the heart rate remains the same as the control case. Aortic valve stenosis does not change the sensitivity to systolic stretch, and regurgitation decreases it, as the value of $\mu_{1}$ is larger than the control case. Similar to the mitral valve, pathologies in the aortic valve do not change the frequency of missed beats.

Sensitivity to diastolic stretch is compared between cases by finding the approximate value of $\mu_{2}$ at which the results begin to bifurcate. Again, a lower value of $\mu_{2}$ is indicative of increased sensitivity. Cases are also compared for similar values of $\mu_{2}$, such that the same level of diastolic stretch is applied. Mitral stenosis slightly reduces sensitivity to arrhythmia stimulating diastolic stretch while aortic stenosis has no effect. Stenosis in either valve does not change the heart rate rise resulting from the MEF. Mitral and aortic valve regurgitation do not affect the sensitivity to diastolic stretch. Furthermore, we find that valve regurgitation increases the heart rate and frequency of ectopic beats resulting from diastolic stretch. This applies whenever $\mu_{2}\neq 0$ and valve regurgitation is introduced. The mechanism behind this result is found to be the increase in myocardial load and diastolic stroke resulting from regurgitation. The diastolic stretch is therefore longer and larger. Arrhythmia (ectopic beats, irregular heart rate) resulting from MEF dysfunction will therefore be more severe. This result agrees very well with clinical results showing that the severity of regurgitation is an indicator for the complications experienced during arrhythmia [47,48,49].

Limitations of the model: The model inherits the same limitations as the model by Kim and Capoccia [28], namely that the 0-dimensional lumped-parameter nature does not allow for wave dynamics in the electrical and cellular behaviour to be modelled. This could limit the investigation of the MEF in which wave dynamics can have a significant effect. Another weakness is the simple electro-chemical model. This model cannot be used to identify the specific ion channels involved in the pattern of cellular excitation.

## 5. Conclusions

This study describes the successful modification and use of the cardiovascular mathematical model developed by Kim and Capoccia [28,32] to study the effects of the pathologies in the mitral and aortic valves. The effects of valve pathologies on the MEF is additionally studied. Mitral valve stenosis increases the sensitivity to arrhythmia-stimulating systolic stretch, but reduces the sensitivity to diastolic stretch. Aortic valve stenosis does not change the sensitivity to arrhythmia-stimulating stretch, and regurgitation reduces it. No significant effect on the sensitivity to diastolic stretch is found for the aorta. A key result is found when valve regurgitation is accompanied by diastolic stretch. Whilst diastolic stretch increases the number of ectopic beats in the presence of MEF disorder, the ectopic beats become far more frequent when accompanied by valve regurgitation. Arrhythmia resulting from a disorder in the MEF will therefore be more severe when valve regurgitation is present, and the more severe the regurgitation the more serious the arrhythmia. This finding agrees well with published clinical literature. A possible mechanism responsible for the rise in the number of ectopic beats is provided.

Finally, we demonstrated how to incorporate valve opening and closing times within our model, indicating some potentially interesting results. It remains a future work to perform detailed analysis on valve motion. It would be also be interesting to extend our study to a continuous model such that the effects of wave dynamics can be studied, particularly in the context of MEF. Extending the electrical model to include a more detailed description of the cellular ion channels is also of interest as the particular channels involved in generating a significant MEF effect can be better studied.

## Figures and Tables

**Figure 1 bioengineering-09-00454-f001:**
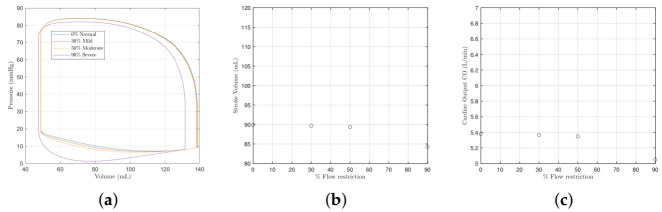
The left ventricle pressure–volume loop, the stroke volume, and the cardiac output with different severities of mitral valve stenosis. The model parameter values used to model each severity are provided in Table 4. (**a**) Pressure–volume loops. (**b**) The stroke volume SV. (**c**) The cardiac output CO.

**Figure 2 bioengineering-09-00454-f002:**
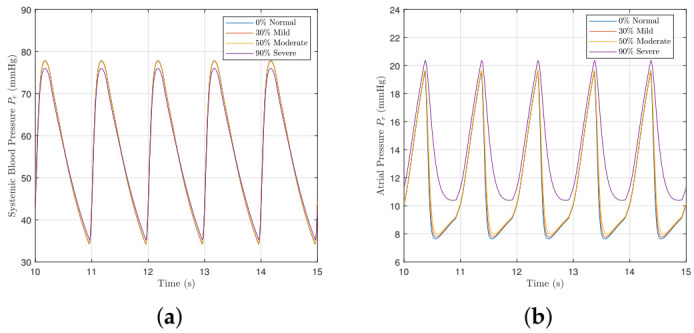
The systemic and atrial pressures with different severities of mitral valve stenosis. (**a**) Systemic blood pressure $P_{s}$. (**b**) Atrial pressure $P_{r}$.

**Figure 3 bioengineering-09-00454-f003:**
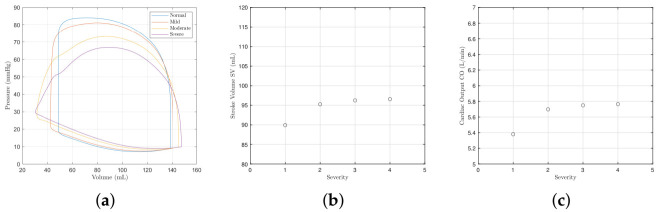
Left ventricle pressure–volume loop, the stroke volume and the cardiac output with different severities of mitral valve regurgitation. The *x* axis of the SV and CO figures give the severity of regurgition, where 1 denotes normal or control, 2 denotes mild, 3 denotes moderate, and 4 denotes severe. The model parameter values used to model each severity are provided in Table 4. (**a**) Pressure–volume loops. (**b**) The stroke volume. (**c**) The cardiac output.

**Figure 4 bioengineering-09-00454-f004:**
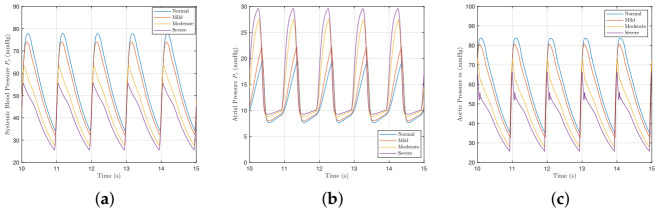
The systemic, atrial, and aortic pressures with different severities of mitral valve regurgitation. (**a**) Systemic blood pressure $P_{s}$. (**b**) Atrial pressure $P_{r}$. (**c**) Aortic pressure *m*.

**Figure 5 bioengineering-09-00454-f005:**
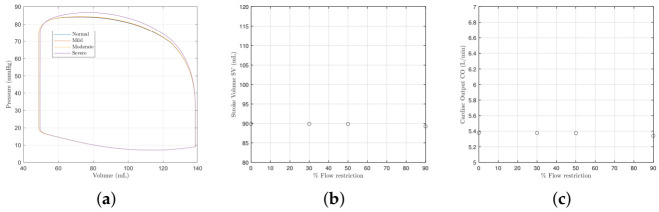
Left ventricle pressure–volume loop, the stroke volume and the cardiac output with different severities of aortic valve stenosis. (**a**) Pressure–volume loops. (**b**) The stroke volume. (**c**) The cardiac output.

**Figure 6 bioengineering-09-00454-f006:**
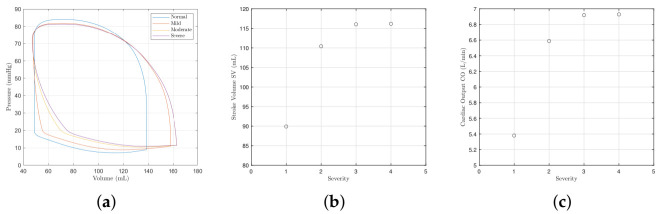
Left ventricle pressure–volume loop, the stroke volume and the cardiac output with different severities of aortic valve regurgitation. The *x* axis of the SV and CO figures give the severity of regurgition, where 1 denotes normal or control, 2 denotes mild, 3 denotes moderate, and 4 denotes severe. The parameter values used to model each severity are provided in Table 4. (**a**) Pressure–volume loops. (**b**) The stroke volume. (**c**) Cardial output.

**Figure 7 bioengineering-09-00454-f007:**
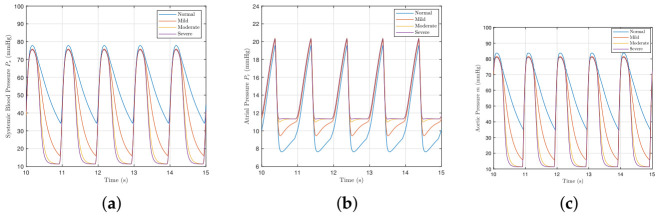
Systemic, atrial, and aortic pressures with aortic valve regurgitation. (**a**) Systemic blood pressure $P_{s}$. (**b**) Atrial pressure $P_{r}$. (**c**) Aortic pressure *m*.

**Figure 8 bioengineering-09-00454-f008:**
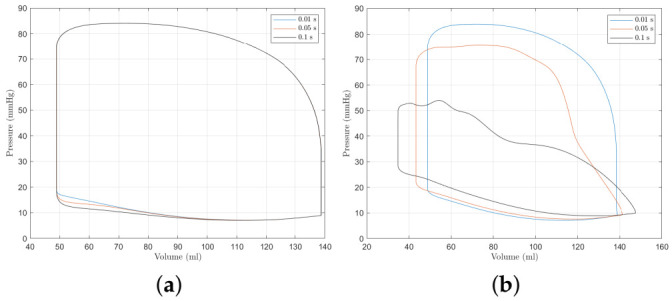
The effect of mitral valve opening and closing time. (**a**,**b**) show, respectively, the effect of opening and closing time on the P-V loop, respectively, and (**c**,**d**) show traces of $U_{mi}$ for different opening and closing times (refer to Section 2.4 for an explanation of $U_{mi}$). (**a**) The effect of mitral valve opening time on the P-V loop. (**b**) The effect of mitral valve closing time on the P-V loop. (**c**) Time traces of $U_{mi}$ for different opening times. (**d**) Time traces of $U_{mi}$ for different closing times.

**Figure 9 bioengineering-09-00454-f009:**
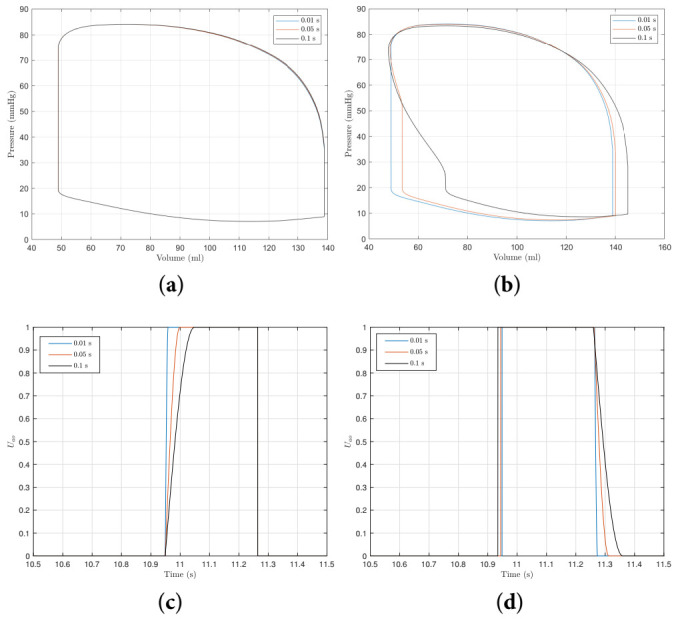
The effect of aortic valve opening and closing time. (**a**,**b**) show, respectively, the effect of aortic valve opening and closing time on the P-V loop, and (**c**,**d**) show traces of $U_{ao}$ for different opening and closing times (refer to Section 2.4 for an explanation of $U_{ao}$). (**a**) The effect of aortic valve opening time on the P-V loop. (**b**) The effect of aortic valve closing time on the P-V loop. (**c**) Time traces of $U_{ao}$ for different opening times. (**d**) Time traces of $U_{ao}$ for different closing times.

**Table 1 bioengineering-09-00454-t001:** Variables and their physiological meaning.

Variable	Physiological Description
$v_{c}$	Velocity of the contractile element (s^−1^)
$\epsilon_{c}$	Strain of the contrile element
$\tau_{c}$	Active tension of the contractile element (mmHg)
$k_{c}$	Stiffness of the contractile element (mmHg)
$\sigma_{p}$	Passive stress (mmHg)
*u*	Chemical activity (s^−1^)
*p*	Slow electric variable
*q*	Fast electic variable
$P_{v}$	Left ventricular pressure (mmHg)
$P_{r}$	Left atrial pressure (mmHg)
$P_{s}$	Systemic pressure (mmHg)
*m*	Aortic pressure (mmHg)
$F_{a}$	Aortic flow (mL/s)

**Table 2 bioengineering-09-00454-t002:** Control parameters for ventricle micro-scale mechanical and electrical models.

Parameter	Value	Physiological Description
$\sigma_{0}$	240 kPa	Maximum left ventricle sarcomere active tension
$k_{0}$	120 kPa	Maximum left ventricle sarcomere active elastance
$k_{1}$,	0.002 kPa	Passive tension parameter
$k_{2}$	14 kPa	Passive tension parameter
χ, $\alpha_{l}$	100 s^−1^, 10 m^−1^	Damping parameters
$\omega_{0}$	100 s^−1^	Microscale oscillation frequency
*a*, *b*	100 m s^−1^ kPa^−1^, 6000 m s^−2^	Active and passive tension parameter
$\beta_{0}$	$\frac{40}{3.5}$ mL^−2^	Frank-starling length-tension parameter
γ	0.6	Left ventricle pressure parameter
$V_{0}$	$\frac{144}{1.5}$ mL	Left ventricle volume parameter
$\alpha_{u}$	5 s^−1^	Ventricle sarcomere chemical excitation parameter
$\mu_{1}$, $\mu_{2}$	0.0024 kPa^−1^, 0 (s mL)^−1^	Ventricle MEF parameters

**Table 3 bioengineering-09-00454-t003:** Circulation parameters.

Parameter	Value	Physiological Description
$R_{a}$	0.001 mmHg s/mL	Aortic valve resistance
$R_{m}$	0.005 mmHg s/mL	Mitral valve resistance
$R_{c}$	0.0398 mmHg s/mL	Characteristic resistance
$R_{s}$	0.5 mmHg s/mL	Systemic resistance
$C_{s}$	1.33 mL/mmHg	Systemic compliance
$C_{a}$	0.08 mL/mmHg	Aortic compliance
$C_{r}$	4.4 mL/mmHg	Atrial compliance
$L_{s}$	0.0005 mmHg s^2^/mL	Inertance of blood in aorta

**Table 4 bioengineering-09-00454-t004:** Mitral and aortic valve pathology values. The different values provided in the table are applied to $U_{mi}$ and $U_{ao}$ to simulate different severities of valve pathologies. Refer to Section 2.4 for an explanation of how this is done. For stenosis, 0.7, 0.5, and 0.1 represent 30, 50, and 90% flow restriction. For regurgitation, the values are identical to [16] who use the clinical definition of regurgitation severity.

Disorder	Mild	Moderate	Severe
Stenosis	0.7	0.5	0.1
Regurgitation	0.004	0.024	0.05

**Table 5 bioengineering-09-00454-t005:** Effect of $\mu_{1}$ with a severe mitral valve stenosis or regurgitation. Refer to Table 4 for severity. $\mu_{1}$ increases from left to right. The headings 2, 4 or 5, 3, and 1 denote the number of P-V loops appearing in the P-V plot as $\mu_{1}$ is increased. The results given in the $\mu_{1}$ rows give the values of $\mu_{1}$ at which the 2, 4 or 5, 3, and 1 P-V loops are observed. SV, CO, and hr rows likewise provide the stroke-volume, cardiac output, and heart rate measured at that value of $\mu_{1}$.

Control	2	5	3	1
μ1	0.0128	0.0145	0.02	0.0268
SV	70.8	70.2	74.0	90.4
CO	4.2	3.7	3.5	2.7
hr	1.0	0.96	0.82	0.50
Stenosis	2	4	3	1
$\mu_{1}$	0.0118	0.0126	0.0190	0.0235
SV	61.1	56.9	71.5	91.1
CO	3.7	3.1	3.3	2.7
hr	1.08	1.12	0.83	0.50
Regurgitation	2	4	3	1
$\mu_{1}$	0.0196	0.0225	0.385	0.055
SV	86.7	75.1	87.5	100.6
CO	5.3	4.3	4.1	3.0
hr	1.03	1.09	0.80	0.50

**Table 6 bioengineering-09-00454-t006:** Effect of $\mu_{1}$ with a severe aortic valve stenosis or regurgitation. Refer to Table 4 for severity. $\mu_{1}$ increases from left to right. The headings 2, 4 or 5, 3, and 1 denote the number of P-V loops appearing in the P-V plot as $\mu_{1}$ is increased. The results given in the $\mu_{1}$ rows give the values of $\mu_{1}$ at which the 2, 4 or 5, 3, and 1 P-V loops are observed. SV, CO, and hr rows likewise provide the stroke-volume, cardiac output, and heart rate measured at that value of $\mu_{1}$.

Control	2	5	3	1
μ1	0.0128	0.0145	0.02	0.0268
SV	70.8	70.2	74.0	90.4
CO	4.2	3.7	3.5	2.7
hr	1.0	0.96	0.82	0.50
Stenosis	2	4	3	1
$\mu_{1}$	0.13	0.0182	0.0220	0.0280
SV	69.6	67.1	70.7	87.8
CO	4.1	3.3	3.3	2.6
hr	0.99	0.92	0.82	0.50
Regurgitation	2	4	3	1
$\mu_{1}$	0.0164	0.0270	0.0315	0.0550
SV	87.4	76.3	77.7	81.1
CO	5.18	3.86	3.59	2.44
hr	1.0	0.90	0.81	0.50

**Table 7 bioengineering-09-00454-t007:** Effect of $\mu_{2}$ with a severe mitral valve stenosis or regurgitation.

μ2	0.153	0.1584	0.1718	0.36	1.8
Control					
SV	78.8	72.3	66.8	45.2	32.3
CO	5.10	5.24	5.45	6.05	6.2
hr	1.04	1.26	1.41	2.25	3.22
Stenosis					
SV	78.3	71.4	64.8	46.6	37.5
CO	4.83	4.85	4.89	5.03	4.8
hr	1.04	1.17	1.32	1.84	2.20
Regurgitation					
SV	75.7	73.1	71.1	57.0	40.8
CO	6.11	6.45	7.40	10.32	11.52
hr	1.49	1.60	1.80	3.02	4.73

**Table 8 bioengineering-09-00454-t008:** Effect of $\mu_{2}$ with a severe aortic valve stenosis or regurgitation.

μ2	0.153	0.1584	0.1718	0.36	1.8
Control					
SV	78.8	72.3	66.8	45.2	32.3
CO	5.10	5.24	5.45	6.05	6.2
hr	1.10	1.26	1.41	2.25	3.22
Stenosis					
SV	81.6	73.0	65.9	45.2	32.0
CO	4.93	5.15	5.30	5.99	6.14
hr	1.01	1.24	1.43	2.22	3.22
Regurgitation					
SV	91.8	86.5	85.5	74.4	45.6
CO	8.67	8.55	9.42	12.89	13.94
hr	1.61	1.68	1.86	2.88	5.10

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
