# Peer review of "An Investigation of Left Ventricular Valve Disorders and the Mechano-Electric Feedback Using a Synergistic Lumped Parameter Cardiovascular Numerical Model"

_bioengineering, 2022, doi:10.3390/bioengineering9090454_

Round 1
Reviewer 1 Report
This manuscript studies a multi-scale compartment model for cardiac electromechanics and the circulatory system. It builds on as series of papers [27,34,35] and uses the same basic multi-scale model for cardiac electrodynamics and its systemic effects. The added novelty here is a simplified representation of valve deficiencies, namely aortic and mitral regurgitation and stenosis. The effects of these valve deficiencies is exhibited by simple parametric senstivitiy studies, and then the question of how these effects influence the mechano-electric feedback is studied.
Comments:
1. The title uses odd punctuation and capitalisation and should be revised.
2. In several places, the model is referred to as being "synergistic" although it is never specific or explained what this means and how exactly the model is synergistic.
3. It is unclear how much a compartmental model without 3-D effects can tell us about MEF. The main critique of this paper is that as a pure modelling paper it does not critically enough assess whether the phenomena are truly reflected in physiology. Some links to electrophysiology studies related to MEF should be added.
4. Page 2, second paragraph: The sentence "In this study a dynamical 0D model..." is grammatically incorrect.
5. Page 3, equations (8-12): Provide a circuit diagram for these equations or a reference to a previous publication containing same.
6. Could the valve model in Section 2.4 be extended to account for gradual opening of the valves? There have been many valve models developed to incorporate aspects like leaflet opening dynamics, see e.g.
Korakianitis T and Shi Y. Numerical simulation of cardiovascular dynamics with healthy and diseased heart valves. J Biomech 39(11), 2006.
7. Tables 1-3: Please provide references for the reference values and if possible a range of physiological values.
8. Figures 8-13: Add the values of mu_1 and mu_2 on each row of these figures to improve their readability.
9. Discussion: I missed the connection with clinical evidence regarding the effect of stenosis and MEF-induced ectopic beats. Have these effects been demonstrated to be present in patients? Are the MEF parameters mu_1 and mu_2 purely theoretical or can they be linked to patient-specific measurements?
In general this manuscript is well-written and presents interesting results regarding a theoretical model for cardiac MEF and the effect of valve deficiencies. The novelty of the model is quite limited as only a simple modification for the reproduction of basic MV/AC pathological effects. This is interesting but not novel compared to many other lumped parameter models. I would have liked to see a deeper exploration of the valve model and more references to known arrhythmic behaviour caused by MV deficiencies. I recommend minor revision to extend the discussion on the points above before the manuscript can be accepted for publication.
Reviewer 2 Report
The paper describes a model of mechano-electric feedback through a lumped parameter model in presence of mitral and aortic valve diseases.
The model is interesting and potentially useful for a better understanding of the effect of valvular diseases on electrical arrhythmias.
The reviewer would strongly recommends the authors to provide a more detailed clinical background giving a better view to the reader about the clinical observations of arrhythmias in presence of valvular diseases.
The other recommendation of the review is to provide some validation of the model. The comparison between the model and clinical/literature data is completely missing. This strongly limits the possibility to claim the model as correct and representative of a clinical phenomenon.
Reviewer 3 Report
The manuscript nicely models the mitral and aortic valve dysfunction and their crosstalk with the MEF severity and consequent rhythm disorder. There exist a few spelling mistakes like " ... As their names name suggests, the MEF is the feedback of local mechanical cell stretch on the electrical ....". The following issues need to be improved:
1- the implementation of the model should be described more in detail. In which environment is it implemented and how the equations have been solved?
2- y axis of stroke volume figures should have the same scales to make the graphs more conclusive by comparing.
3- Figures 8 to 15 must be analyzed for example by stroke volume, and they could be presented like Fig6b in a way that all combinations of valve dysfunction and MEF disturbances will be represented in one chart. In this way, Figures 8 to 15 could be summarized more comprehensively and fewer in number. (p, q irregularities could be analyzed in the same way)
4- It makes sense to discuss a few sentences about the so-called results of published literature! " ... These results match well with published literature [37, 14, 36] "
Round 2
Reviewer 2 Report
As I stated in the first revision, the paper for me is not of enough quality for publication. The amount of revision needed from my side (and in turn author side) to improve a paper to a level suitable for publication goes beyond a reasonable time commitment for a revision. The validation conducted lacks the hemodynamic aspect, and that is a critical point when developing an electrico-mechanical model of the heart. And it should be conducted in a quantitative way on variables and not in general and qualitative terms (e.g. data are in agreement with author 1, author 2 etc.). For this reason I discourage publication